# Deformable Linear Object Manipulations with Differentiable Physics

## Abstract

We address the challenge of enabling robots to manipulate deformable linear objects (DLOs), such as wires, ropes, and rubber bands. Prior work in this domain has primarily focused on narrow, task-specific problems, often relying on real-world demonstrations or handcrafted heuristics. Such approaches, however, do not scale to the diverse range of materials and tasks encountered in practice, where collecting sufficiently varied real-world data is impractical. Moreover, existing simulation environments provide limited support for the broad spectrum of material behaviors required for generalizable DLO manipulation. To overcome these limitations, we introduce a differentiable physics simulator specifically designed for versatile DLO manipulation. Our simulator models a wide range of material properties—including extensibility, inextensibility, elasticity, bending plasticity, and interactions with both rigid and deformable objects—thereby establishing a robust foundation for learning and evaluating manipulation skills. Building on this simulator, we propose a benchmark suite of representative DLO manipulation tasks that highlight their unique challenges. We further evaluate multiple policy learning algorithms on these tasks. The results show that reinforcement learning can learn closed-loop policies but requires prohibitively large amounts of data. In contrast, trajectory optimization is more efficient: gradient-based methods achieve the best sample efficiency when gradients are available, while sampling-based approaches are broadly applicable but less efficient. Please refer to the supplementary material for video demonstrations.

## 1 Introduction

Deformable object manipulation has long been a challenging topic in the field of robotic manipulation (Sanchez et al., 2018; Yin et al., 2021; Zhu et al., 2022). Among the diverse forms of common deformable materials, deformable linear objects (DLOs), such as wires and ropes in Figure 1(a), exhibit unique features and properties. Compared to volumetric materials, a DLO can be represented by a relatively small set of vertices and degrees of freedom due to its linear co-dimensional nature. However, this apparent simplicity does not make manipulation any easier. DLO manipulation introduces several unique challenges: (1) high-precision manipulation is required due to the co-dimensional nature of the object, (2) DLOs exhibit complex dynamics that are difficult to model, and (3) perception algorithms often struggle to perceive the state of DLOs when self-crossings occur.

Prior work has addressed DLO manipulation in several directions. (1) Approaches focusing on perception, such as cable-tracing tasks (Caporali et al., 2022; Kicki et al., 2023; Shivakumar et al., 2022; Viswanath et al., 2023), which require extensive data collection and rely heavily on synthetic datasets for training. (2) Approaches focusing on manipulation, such as cable untangling (Viswanath et al., 2021; Grannen et al., 2020; Sundaresan et al., 2021) and shape control (Yu et al., 2022a;b; Caporali et al., 2024), which generally target specific tasks and are difficult to generalize to other DLO-related problems. From this perspective, we argue that a simulation and benchmark platform for DLO manipulation is necessary—both for generating perception training data and for facilitating policy learning for manipulation.

Several prior works have proposed simulations for modeling DLO dynamics (Lin et al., 2021; Naughton et al., 2021; Li et al., 2021; Tong et al., 2023; Chen et al., 2024). However, as shown in Table 1, each of them only supports a subset of the desired features. Among these features, cou-

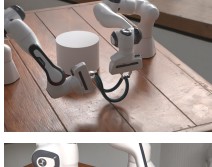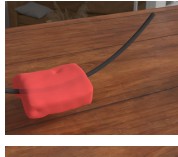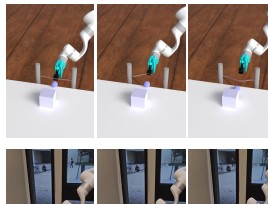

(a) Common DLOs in daily life     (b) Manipulation benchmark     (c) Coupling     (d) Real-world experiment

Figure 1: **DLO-Lab.** (a) We encounter various deformable linear objects (DLOs) in our daily life, such as cables, ropes, and wires. (b) To facilitate versatile robotic skill learning for DLOs with different material properties, we introduce *DLO-Lab*, a fully differentiable simulation environment accompanied by a set of benchmark tasks. (c) Our simulator effectively supports coupling with other materials, enabling the interaction between DLOs and other objects. (d) We also conduct real experiments to deploy a trained policy in the real world.

pling and differentiability are the two most important properties to enable flexible manipulation and efficient policy optimization. On one hand, coupling effectively models the physical interactions between DLOs, robot manipulators, and other objects in the scene. This physical fidelity is the foundation for simulating a wide range of complex, contact-rich manipulation tasks. On the other hand, differentiability is crucial for efficient policy optimization. It provides gradient information of the system's dynamics, which quantifies how changes in the robot's actions influence the final state of the DLO. However, none of the previous studies address *both* coupling and differentiability, and they also overlook other common properties found in DLOs, such as bending plasticity and loop topology. Therefore, we identify the need for a simulation environment that integrates diverse material couplers and is tailored for robotic skill acquisition in various DLO manipulation tasks.

To this end, we propose a differentiable simulation engine, DLO-Lab, featuring diverse material behaviors of DLOs and their interactions with other entities. Building upon the Genesis platform (Authors, 2024), we implement a solver for DLOs and their coupling with various materials, including rigid bodies, deformable objects, and beyond. Most operators in our solver are naturally differentiable by leveraging Taichi (Hu et al., 2019; 2020) as the programming language. Specifically, our DLO solver is based on (Bergou et al., 2008; 2010), and we further enrich it with bending plasticity and support for loop topologies. In addition, we implement frictional contact between DLOs and other materials, and also ensure the differentiability requirement during contacts. As a result, our simulation is able to simulate the coupling between DLOs with other materials supported by Genesis, such as fluid and elastic bodies, as shown in Figure 1(c).

With this simulation, we design a set of benchmark tasks that highlight the properties and capabilities of DLOs. We evaluate various algorithms, including reinforcement learning methods (Schulman et al., 2017; Haarnoja et al., 2018), sample-based trajectory optimization (Hansen & Ostermeier, 2001), and gradient-based approaches that directly benefit from differentiable simulation. Some DLO manipulation tasks are typically long-horizon in nature, requiring multiple re-grasping steps as shown in Figure 1(b). To address this, we introduce an agent pipeline that decomposes tasks into atomic subtasks. Leveraging the linear structure of DLOs, we can represent the grasping point on a rope by a single value, prompting the agent to output the grasping point given the task.

Our experiments suggest that gradient-based methods achieve the highest sample efficiency when gradients are available, while sampling-based approaches are more broadly applicable but less efficient. Reinforcement learning algorithms can, in principle, learn closed-loop policies, but require many more samples to obtain a successful trajectory. To further examine the sim-to-real gap, we conduct real-world experiments with open-loop manipulation in Figure 1(d). The results show that our simulation maintains a reasonable level of realism, enabling tasks to be transferred effectively.

We summarize our main contributions as follows:

• We propose a differentiable simulation engine for DLOs and their interactions with various materials. Our simulation environment accommodates a wide range of material configurations and offers a standardized interface suitable for both RL-based and gradient-based policy learning algorithms.

- We introduce DLO-Lab, a comprehensive benchmark for robotic manipulation involving different types of DLOs. This benchmark aims to facilitate the exploration of versatile manipulation skills across a diverse set of DLOs. Using DLO-Lab, we analyze the performance of sampling-based and gradient-based optimization methods, as well as the challenges they encounter.

- We propose an agent framework to tackle the challenges of long-horizon DLO manipulation, leveraging the linear properties of DLOs to effectively define the grasp point for each sub-task once the gripper has been released in previous stages.

## 2 RELATED WORK

**Differentiable Simulation**    refers to computational frameworks where physical dynamics are fully differentiable with respect to system and control parameters. This paradigm has gained significant popularity in robotics and computer graphics, allowing for efficient, gradient-based optimization of policies (Mora et al., 2021; Huang et al., 2021; Xu et al., 2022; Xian et al., 2023; Wang et al., 2024) and system parameters (Li et al., 2022a; Xue et al., 2023; Ma et al., 2023; Cao et al., 2024; Lin et al., 2025) by back-propagating directly through the simulation. In general, these frameworks can be divided into two main categories. *Learning-based approaches* use neural networks to approximate the forward dynamics (Sanchez-Gonzalez et al., 2020; Pfaff et al., 2021; Chen et al., 2024) without explicit formulations. These methods are naturally differentiable through the networks and are typically trained on ground-truth simulation data to predict state transitions. Despite their efficiency, learning-based methods require substantial training data and often have limited generalizability. *Analytical models*, on the other hand, adhere to classic simulation methods, *e.g.*, finite element methods (FEM) (Sifakis & Barbic, 2012) and material point methods (MPM) (Jiang et al., 2016), and calculate exact gradients of the underlying physical models. Some previous works, like DiffTaichi (Hu et al., 2020) and Warp (Macklin, 2022), use automatic differentiation with explicit time-stepping schemes. Others, such as DiffSim (Qiao et al., 2020) and DiffCloth (Li et al., 2022b), derive analytical gradients and employ iterative solvers for implicit time integration.

**Deformable Linear Object Manipulation**    represents a significant and long-standing challenge in robotics, with critical applications in industrial assembly, surgical robotics, and household tasks (Saha & Isto, 2006; Lee et al., 2021; Laezza & Karayiannidis, 2021; Yu et al., 2022a; Lv et al., 2022; Zhaole et al., 2024). Existing literature broadly categorizes DLO manipulation into two paradigms (Laezza & Karayiannidis, 2021; Laezza et al., 2021; Monguzzi, 2023). The first, explicit shape control, aims to deform the DLO into a precise geometric configuration. Works in this area often employ neural networks to learn the state transition of DLO dynamics, aiming to achieve a designated shape (Yan et al., 2020; Wang et al., 2022; Chen et al., 2024). In contrast, implicit shape control focuses on fulfilling high-level task conditions where the DLO's exact shape is not the primary objective. Such tasks include tying knots (Saha & Isto, 2006), routing cables (Luo et al., 2024), and fixing cables into clips (Li et al., 2018). Although these methods have achieved success in real-world applications, they are typically task-specific and difficult to generalize. Our work addresses this limitation by introducing a comprehensive simulation environment designed to facilitate the learning of versatile DLO manipulation policies.

## 3 SIMULATION ENVIRONMENT

In this work, we introduce a fully differentiable simulation environment tailored for DLO manipulation. We implemented the simulator using the Taichi (Hu et al., 2019; 2020) programming language and integrated it into Genesis (Authors, 2024) to build a versatile benchmark for DLO manipulation tasks. Our simulator models a broad range of DLO's behaviors and its interactions with other rigid or soft bodies. Table 1 shows a comparison between our simulation engine and other relevant works for DLO modeling, including Bi-LSTM (Yan et al., 2020), GNN (Wang et al., 2022), XPBD (Liu et al., 2023), SoftGym (Lin et al., 2021), Elastica (Naughton et al., 2021), C-IPC (Li et al., 2021), IMC (Tong et al., 2023), and DEFORM (Chen et al., 2024).

| Simulators | Elastic Potentials | Bend. Plas. | Loop Topo. | Coupling | Diff. | Skill Learning |
|---|---|---|---|---|---|---|
| Bi-LSTM | | | | | ✓ | |
| GNN | | | | | ✓ | |
| XPBD | | | | | ✓ | |
| SoftGym | | | | ✓ | | ✓ |
| Elastica | ✓ | | | ✓ | | |
| C-IPC | ✓ | | | ✓ | | |
| IMC | ✓ | | | | | |
| DEFORM | ✓ | | | | ✓ | |
| AGX Dynamics | ✓ | ✓ | | ✓ | | |
| Ours | ✓ | ✓ | ✓ | ✓ | ✓ | ✓ |

Table 1: **Comparison with existing simulators for DLOs.**

## 3.1 DLO Modeling

Following the classic discrete elastic rod (Bergou et al., 2008; 2010) methods, DLO-Lab represents a deformable linear object $\mathbf{X}$ as a centerline specified by a set of $N_v$ vertices and a set of $N_e$ adapted, orthonormal frames attached to each edge:

$$\mathbf{x} = \{\mathbf{x}_i | \mathbf{x}_i \in \mathbb{R}^3\} \text{ and } \mathbf{d} = \left\{(\mathbf{d}_1, \mathbf{d}_2, \mathbf{d}_3)^j | \mathbf{d}_1, \mathbf{d}_2, \mathbf{d}_3 \in SO(3)\right\}, \quad (1)$$

where $0 \leq i \leq N_v$ and $0 \leq j \leq N_e$. The term *adapted* means that $\mathbf{d}_3^j$ lies along the edge given by the adjacent vertices, $\mathbf{e}^j = \mathbf{x}_{j+1} - \mathbf{x}_j$, and $\mathbf{d}_3^j \equiv \mathbf{e}^j/|\mathbf{e}^j|$. During forward simulation, the internal interaction of DLOs is modeled by the potential energy $U(\mathbf{X}^t)$, where $\mathbf{X}^t$ denotes the state variables (*i.e.*, vertex states $\mathbf{x}^t$ and frame states $\mathbf{d}^t$) of the DLO at time $t$. Concretely, the potential energy is composed of stretching energy $U_s(\mathbf{d}^t)$, bending energy $U_b(\mathbf{x}^t)$, and twisting energy $U_t(\mathbf{x}^t)$. We calculate the derivatives $\partial U/\partial \mathbf{X}^t$ and employ a standard symplectic Euler solver for time stepping. In addition, we model the bending plasticity by introducing a yield threshold $\sigma_y$ and a creep rate $r_c$, which are used to adjust the rest curvatures when the yield constraint is violated. Furthermore, we model the frictional contact behavior between DLOs using position-based dynamics (Müller et al., 2007), which simplifies the process of solving complex and computationally expensive non-linear equations typically required by preventative methods like IPC (Li et al., 2020; 2021). Please refer to Appendix A.1 for more details about the representation of DLOs used in this work.

## 3.2 Coupling Implementation

**Rigid Body** We adopt the rigid solver from Genesis and implement a two-way coupling scheme between DLOs and rigid bodies. The rigid solver models each rigid geometry as a time-varying signed distance field (SDF). At each simulation step, we calculate the penetration depth $d$ of each rigid geometry for every sampled position $\mathbf{p}$ along the centerline of the DLO: $d(\mathbf{p}) = r(\mathbf{p}) - \text{SDF}(\mathbf{p})$, where $r$ is the radius. Rather than a hard binary contact, we use a soft-coupling approach where an exponential influence factor $f_i$, determined by a tunable softness parameter $\epsilon_s$, is calculated based on the penetration depth: $f_i = \min\left(\exp(d/\epsilon_s), 1\right)$. When $f_i$ exceeds a predefined threshold (0.1 as used in this work), we resolve the collision using an impulse-based response. The contact normal is derived from the gradient of the SDF, and we decompose the relative velocity between the sampled location of DLO and the contact point on the rigid body into normal and tangent components. This allows us to apply a standard restitution model to the normal component and a friction model to the tangential component. We calculate the change in the momentum at $\mathbf{p}$ and apply an equal and opposite reaction force back to the rigid body to ensure momentum conservation. Additionally, to stabilize the grasp with a rigid gripper, we identify the nearest vertices in contact with the manipulator and flag them as kinematic for the current time step. This prevents the DLO's internal position-based collision handling scheme from generating conflicting updates, ensuring that the DLO deforms compliantly around the gripper.

**Soft Body** To simulate the complex interactions between DLOs and other volumetric materials, such as fluids or soft bodies, we propose a two-way coupling scheme between our DLO solver and the Material Point Method (MPM) (Jiang et al., 2016; Hu et al., 2018) solver implemented in Genesis. This coupling is facilitated through the background Eluerian grid employed by MPM. Specifically, in the first direction from the DLO solver to the MPM solver, we treat the discrete vertices of

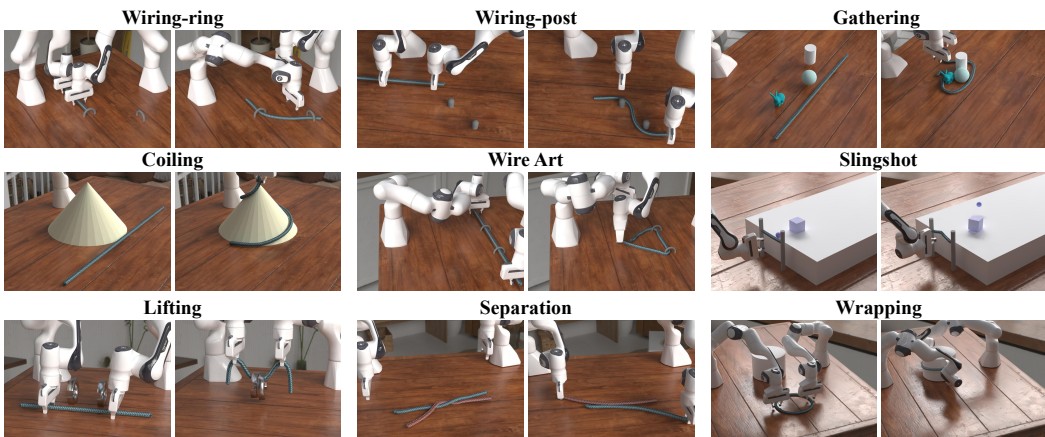

Figure 2: **Task illustration.** We show the manipulation tasks in our benchmark, each with an initial state and the desired goal state.

the DLO as Lagrangian markers. During the particle-to-grid transfer (P2G) step of MPM, the mass and momentum of each vertex are rasterized onto the surrounding grid nodes. This contributes to the grid's momentum field in conjunction with the MPM particles, enabling the DLO to influence the motion of the MPM material directly. In the second direction, from MPM back to the DLO, forces are transferred after the grid update step. For each DLO vertex, a change in momentum is interpolated from the surrounding grid nodes. This change is then applied as an impulse within our DLO solver, enabling the MPM material to affect the DLO. This staggered, grid-based approach ensures stable and physically plausible two-way coupling, allowing us to simulate rich multi-material phenomena, as shown in Figure 1(c).

## 3.3 GRADIENT BACK-PROPAGATION

Our simulator, implemented with Taichi (Hu et al., 2019; 2020), benefits from its automatic differential tool (autodiff) to calculate numerical gradients. While this method is effective for most of our simulations, standard autodiff can become numerically unstable when dealing with frictional contact. The main issue arises from gradient explosion, which occurs due to division by near-zero distances when calculating the contact normal. To mitigate this problem, we implement custom back-propagation kernels specifically designed for handling self-collision and friction. We adopt a "frozen normal" approximation, which involves treating the normal from the forward pass as a constant during gradient calculations. Additionally, we use ratio-preserving norm clipping to limit the gradient magnitude, further stabilizing the back-propagation of gradients. Moreover, we also implement gradient checkpointing to enable gradient back-propagation through long horizons. More details can be found in Appendix A.2.

## 4 DLO-LAB BENCHMARK

Built upon the proposed differentiable simulation engine, our DLO-Lab benchmark provides a versatile collection of deformable linear object manipulation tasks equipped with differentiable reward functions. We also offered standardized APIs for building new manipulation environments and learning different policies. Our proposed tasks are illustrated in Figure 2.

### 4.1 BENCHMARK REPRESENTATION

**Task Formulation**  We address a series of manipulation tasks where an agent operates a set of robot grippers to achieve a specific goal. We formulate each task as a standard finite-horizon Markov Decision Process (MDP). Specifically, an MDP is comprised of a state space $\mathcal{S}$, an action space $\mathcal{A}$, a transition function $\mathcal{T} : \mathcal{S} \times \mathcal{A} \rightarrow \mathcal{S}$, and a reward function associated with each transition step $\mathcal{R} : \mathcal{S} \times \mathcal{A} \times \mathcal{S} \rightarrow \mathbb{R}$. We optimize the agent to derive a policy $\pi(a|s)$ that maximizes the expected sum of discounted rewards, $E_\pi\left[\sum_{t=0}^{T} \gamma^t \mathcal{R}(s_t, a_t)\right]$, over a horizon $T$ with a discount factor $\gamma$.

**State Space**   Let $N_v$ denote the total number of vertices of all DLOs in the scene, and let $N_m$ denote the total degrees of freedom (DoF) for all robot arms. The complete simulation state for the system is represented as:

$$\mathbf{S} = (\mathbf{x}, \dot{\mathbf{x}}, \mathbf{r}, \mathbf{M}, \dot{\mathbf{M}}),$$

where $\mathbf{x}, \dot{\mathbf{x}} \in \mathbb{R}^{N_v \times 3}$ gives the positions and velocities of all DLO vertices, $\mathbf{r} \in \mathbb{R}^{N_v}$ encodes the rest configurations, and $\mathbf{M}, \dot{\mathbf{M}} \in \mathbb{R}^{N_m}$ denote the joint positions and velocities of the robot arms.

**Observation**   Unlike other deformable-material manipulation settings (Huang et al., 2021; Xian et al., 2023; Wang et al., 2024) where downsampling is required to reduce the dimensionality of the observation space, in our case, it is computationally feasible to leverage the full state of the DLOs. Thus, the observation space for the DLOs consists of the stacked positions and velocities $(\mathbf{x}, \dot{\mathbf{x}})$, resulting in an $N_v \times 6$ representation. To capture the robot side of the interaction, we augment the observation with each manipulator's end-effector position $p_i$ and orientation $r_i$, along with the current joint configuration $\mathbf{M}$. This yields a complete observation vector that provides the agent with direct access to both the DLO dynamics and the manipulator kinematics, ensuring sufficient information for closed-loop control.

**Action Space**   We apply position-based control to the robot arms. At each step, the action corresponds to the target joint positions of all robot arms, represented as desired values for their respective DoFs. This formulation provides direct control authority at the kinematic level, allowing the policy to specify joint configurations without requiring additional inverse kinematics computation. As a result, the control signal is both simple and expressive, enabling precise manipulation of the DLOs through coordinated arm motion.

### 4.2 Manipulation Tasks

**Wiring**   This task requires the agent to operate two robot arms to route the wire along the designated track. Based on the widgets for specifying the track, we split the task into two sub-tasks: **Wiring-ring** and **Wiring-post**.

**Gathering**   This task starts with several rigid and deformable bodies randomly placed on the ground. The objective is to use a rope to gather these objects together.

**Coiling**   The scene consists of a cone and a rope. The agent needs to wind the rope around the surface of the cone.

**Wire Art**   Given a wire with bending plasticity, the agent is tasked with bending the wire into a target triangular shape.

**Slingshot**   In this scenario, a rigid ball and a rigid cube are placed on a table. In front of the ball, there is a slingshot made from a rope with high stretching stiffness. The goal is for the agent to operate a robot arm to use the slingshot to launch the ball and hit the cube.

**Lifting**   Given two "C"-shaped rings, the agent manipulates a rope to lift the rings and make them come into contact.

**Separation**   In this task, the agent needs to use two robot arms to separate two ropes that are initially tangled together.

**Wrapping**   In this task, the agent utilizes two robot arms to wrap a rubber band around a cylinder with a radius slightly larger than that of the rubber band.

Please refer to Appendix B.1, where we provide more details about task setups.

### 4.3 Reward Design

Here, we summarize the reward design used for our manipulation tasks.

In the **Wiring-ring** task, we define the reward to be the sum of three parts: (1) We encourage the rope to get close to the center of the rings, and calculate the sum of the minimum distances from the rope to the centers of the rings. The negative value is added to the reward. (2) We also encourage the rope to point through the rings, so we calculate $\sigma(\mathbf{v} \cdot \mathbf{n})$, where $\sigma(\cdot)$ is the sigmoid function, $\mathbf{n}$ is the ring's normal, and $\mathbf{v}$ is the moving direction of the rope segment with minimum distance to the center of the ring. (3) We also define a smooth curve connecting the centers of the two rings, and calculate the negative distance between the rope and the curve.

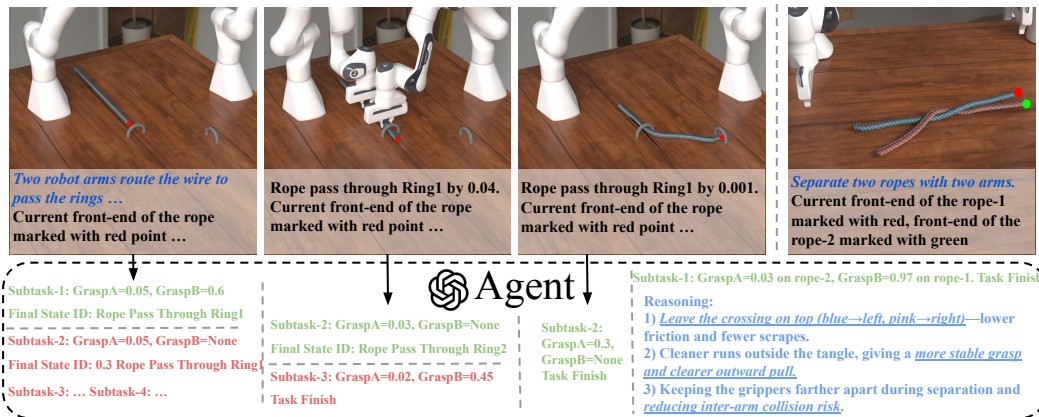

Figure 3: We propose an agent system that decomposes long-horizon tasks and identifies reasonable grasping points for manipulation. To improve accuracy, the process is iteratively updated: after completing each subtask, the agent reevaluates the current state and generates new subtasks based on actual conditions rather than imagined outcomes. Moreover, we show that the agent is able to reason effectively about both morphological structure and physical properties of the objects.

For the **Wiring-post** task, the reward is defined as the sum of two parts: the negative total of the minimum distances from the rope to one side of each post that should make contact, and the negative distance from the rope to a predefined curve that connects the two posts.

For **Gathering**, the reward is defined as the sum of the negative distances between the center of mass for each pair of objects.

In the **Coiling** task, we define the reward as the sum of the negative distances from each vertex of the DLO to the center of the cone.

For **Wire Art**, since a target shape is defined, we set the reward as the negative distance between the current position and the target position.

For **Slingshot**, the reward is defined as the total moving distance of the cube projected onto the $+y$-axis (the direction away from the slingshot) in 500 time steps after the slingshot is released.

For **Lifting**, we define the reward as the negative distance between the two "C-shaped" rings, along with the sum of their heights from the ground.

For **Separation**, we calculate the chamfer distance between the two ropes and use it as the reward.

Finally, for **Wrapping**, we uniformly sample 20 points around the bottom circle of the cylinder and calculate the maximum distance between a sampled point from the cylinder and a vertex from the rubber band. We take the negative value as the reward.

The loss used for gradient-based methods is generally negative rewards in those tasks, while we also apply gradient norm clipping to stabilize the training.

## 4.4 AGENT TASK DECOMPOSITION

In deformable linear object (DLO) manipulation, the choice of grasping point is critical for successful execution. For tasks such as **Wire Art**, **Coiling**, **Lifting**, and **Gathering**, effective grasps typically lie at the ends of the object. In **Slingshot** and **Wrapping**, however, contact-point selection is more intricate and depends on the object's dynamics and physical properties. The **Wiring** task is more demanding still: it requires switching grasp points multiple times; without such switches, the task is infeasible. Manually specifying every contact point is tedious, and blindly sampling candidates via trial-and-error is computationally expensive.

We address this by leveraging common-sense knowledge from foundation models to (1) decompose complex tasks into multi-step subtasks and (2) propose a grasping point for each subtask. As illustrated in Figure 3, we prompt the agent with the task description, a rendering of the initial configuration, and markers indicating the rope's front end. Owing to the rope's linear topology, any

| Tasks | Wiring-ring | Wiring-post | Gathering | Coiling |
|---|---|---|---|---|
| PPO | -0.8003±0.0008 | -0.9822±0.0050 | -1.3934±0.0518 | -50.5882±0.3242 |
| SAC | -0.3552±0.0002 | -0.4877±0.0001 | -1.2103±0.0037 | -23.2357±0.2762 |
| CMA-ES | -0.1084±0.0036 | **-0.0893±0.0091** | **-0.5722±0.0123** | -8.0678±0.1302 |
| GD | **-0.0329±0.0011** | -0.0975±0.0008 | – | **-6.8980±0.3426** |

| Tasks | Wire Art | Slingshot | Lifting | Separation | Wrapping |
|---|---|---|---|---|---|
| PPO | -0.5282±0.0031 | 2.1999±1.6525 | -0.0899±0.0031 | **7.3491±0.8033** | -0.8983±0.0034 |
| SAC | -0.2082±0.0001 | 0.9643±0.0260 | -0.0950±0.0573 | 0.2945±0.0113 | -0.5549±0.0249 |
| CMA-ES | -0.0234±0.0041 | **5.0000±0.0000** | **-0.0235±0.0059** | 1.5469±0.0126 | -0.1480±0.0025 |
| GD | **-0.0084±0.0006** | – | – | 1.3600±0.0110 | **-0.1194±0.0007** |

Table 2: **Benchmark results.** We show the maximum reward within a fixed number of episodes and its standard deviation.

point on it can be parameterized by a scalar $s \in [0, 1]$, which allows the agent to output a grasp as a single float specifying the grasp location.

Relying solely on open-loop *imagination* from a large model can introduce inconsistencies across steps (*e.g.*, Figure 3 shows that reusing a previous subtask's grasp may fail). To mitigate this, after each subtask, we re-run the task-decomposition procedure using the updated state. This closed-loop re-planning significantly improves success rates and reduces planning errors in subsequent steps.

We also enable the agent to explain why it selects particular contact points. As shown in Figure 3, the agent reasons correctly in **Separation** by choosing two contact points that most efficiently complete the task. Its rationale reflects awareness of physical properties such as friction and rope topology, suggesting that foundation models possess a rudimentary intuitive understanding of such tasks.

## 5 EXPERIMENT

Based on our benchmark tasks, we quantitatively evaluate methods for state-based policy learning and trajectory optimization algorithms, including Reinforcement Learning algorithms, sample-based trajectory optimization, and, thanks to the differentiable simulator, we can evaluate gradient-based trajectory optimization. We also conduct real-world experiments to verify the effectiveness of sim-to-real transfer.

### 5.1 BENCHMARKED METHOD

We conduct a comprehensive performance evaluation of various methods. For Reinforcement Learning (RL) algorithms, we evaluate Soft Actor-Critic (Haarnoja et al., 2018) and Proximal Policy Optimization (Schulman et al., 2017). For the sample-based trajectory optimization method, we use Covariance Matrix Adaptation Evolution Strategy, namely CMA-ES (Hansen & Ostermeier, 2001). For gradient-based trajectory optimization, we directly apply the gradient to the trajectory and perform gradient descent optimization using the Adam optimizer (Kingma, 2014).

### 5.2 RESULT AND ANALYSIS

**Reinforcement Learning Algorithms.** Across most tasks, reinforcement learning (RL) algorithms perform significantly worse than trajectory optimization methods when given the same number of sampled trajectories (as shown in Table 2). This is expected, as in our setting, the evaluation is based on the highest score achieved during sampling. RL, which aims to learn a closed-loop policy, carries a greater burden: the policy network requires a warm-up phase and more state-space exploration before meaningful learning occurs. As a result, RL tends to be less sample-efficient than open-loop policy learning methods in this context.

**CMA-ES.** The sample-based trajectory optimization algorithm CMA-ES achieves the strongest overall performance, as shown in both Table 2 and Figure 4. This can be attributed to two factors: (1) it does not target a closed-loop policy network, and (2) it does not rely on gradient information,

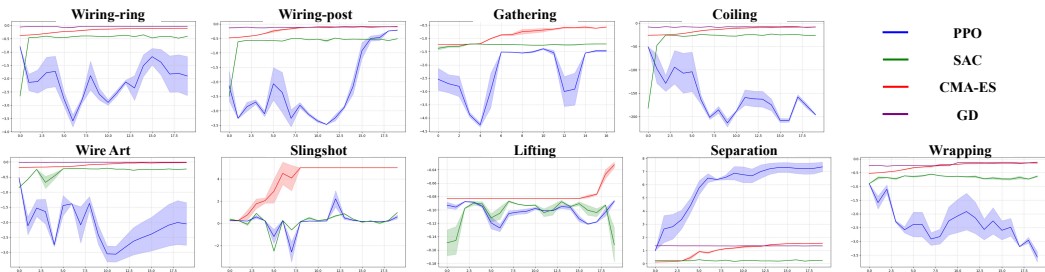

Figure 4: **Reward curve comparisons.** We report the average reward (the vertical axis) with respect to the number of training episodes (the horizontal axis) for different approaches.

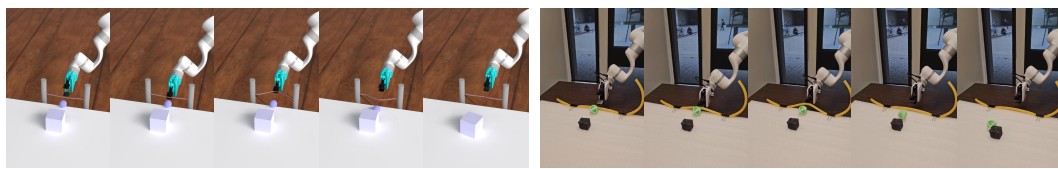

Figure 5: **Sim-to-Real transfer results.**

which is often difficult or impossible to obtain in certain tasks (*e.g.*, **Lifting**, **Gathering**, and **Slingshot**). Furthermore, CMA-ES can leverage massive parallelization in simulation to accelerate the sampling process. Although it may not match the theoretical sample efficiency of gradient-based methods, in practice, it can reach competitive performance more quickly, making it an attractive choice in complex settings.

**Gradient-based Trajectory Optimization.** A central limitation of gradient-based trajectory optimization is the inaccessibility of gradients in some tasks. For example, in **Lifting**, **Gathering**, and **Slingshot**, the final reward depends on the positions of rigid objects. The gradient signal must be propagated through contact interactions with the rope or manipulator, but in an initial trajectory, such gradients may not exist, making the optimization intractable. However, in tasks where gradients are available, this method achieves the highest sample efficiency among all baselines. This demonstrates both the effectiveness of gradient-based optimization and the importance of differentiable simulation when solving contact-rich tasks.

### 5.3 REAL-WORLD EXPERIMENT

We conduct a real-world experiment on the **Slingshot** task using a single robot arm. To demonstrate that our simulation exhibits a reasonable sim-to-real gap, we train an open-loop policy via sample-based trajectory optimization in simulation and transfer it, without additional tuning, to a real-world digital-twin setting. The results in Figure 5 show that the action trajectory that succeeds in simulation can also be executed on hardware and achieves similar outcomes in the physical environment, indicating that, in this setting, the sim-to-real gap can be effectively bridged.

## 6 CONCLUSION

We introduced *DLO-Lab*, a differentiable simulation engine and benchmark for manipulating deformable linear objects, combining a discrete-rod–based solver—with bending plasticity, loop topologies, and frictional contact—and a suite of representative tasks that capture the long-horizon, contact-rich nature of DLO manipulation. Exploiting the unique property that each point on a DLO can be mapped to a one-dimensional coordinate $s \in [0, 1]$, we build an agent system that decomposes long-horizon tasks and proposes grasp points accordingly. Using these selected grasp points, we benchmark multiple policy-learning algorithms—including reinforcement learning, sample-based trajectory optimization, and gradient-based methods—and find, through a real-world experiment, that the sim-to-real gap is manageable: open-loop policies transferred from simulation achieve similar outcomes on hardware. We believe DLO-Lab provides a practical foundation for advancing learning-based DLO manipulation, inviting future work on richer material models, perception-in-the-loop control, and broader sim-to-real validation.

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

# A  IMPLEMENTATION DETAILS

## A.1  DLO REPRESENTATION

In this work, we follow discrete elastic rod (Bergou et al., 2008; 2010) methods to model deformable linear objects (DLOs). Concretely, a DLO $\mathbf{X}$ is a centerline given by a set of $N_v$ vertices $\{\mathbf{x}_i\}_{i=1}^{N_v}$ and a set of $N_e$ orthonormal material frames $\{(\mathbf{d}_1, \mathbf{d}_2, \mathbf{d}_3)^j\}_{j=1}^{N_e}$ attached to each edge $\mathbf{e}^j = \mathbf{x}_{j+1} - \mathbf{x}_j$. Each vertex has its mass $m_i$ and radius $r_i$. Note that we use lower and upper indices to specify vertex- and edge-based properties, respectively. We assume $\mathbf{d}_3^j$ is adapted to the tangent $\mathbf{t}^j$ to the centerline, *i.e.*, $\mathbf{d}_3^j \equiv \mathbf{t}^j = \mathbf{e}^j/|\mathbf{e}^j|$. At the initial time $t = 0$, we predefine a reference frame $(\underline{\mathbf{d}}_1, \underline{\mathbf{d}}_2, \underline{\mathbf{d}}_3)$. The material frame can be obtained at a later time through a twist angle $\theta^j$ with respect to the reference frame. Thus, the total degrees of freedom (DoF) associated with the DLO, given by $N_v$ vertices and $N_e$ twist angles, is $3 \times N_v + N_e$.

To achieve elastic simulation for a DLO, we need to compute its elastic potential as defined by strain. Based on Kirchhoff's rod model (Dill, 1992; O'Reilly, 2017), strains can be separated into three categories: stretching, bending, and twisting.

• **Stretching** The stretching strain, or axial strain of an edge $\mathbf{e}^j$ is given by:

$$\epsilon^j = \frac{|\mathbf{e}^j|}{|\bar{\mathbf{e}}^j|} - 1, \tag{2}$$

where $|\bar{\mathbf{e}}^j|$ is the rest edge length. Denote $k_s^j = KA^j$ as the stretching stiffness, where $K$ is the stretching modulus and $A^j$ is the cross-sectional area, the stretching energy is then defined as:

$$U_s = \frac{1}{2} \sum_{j=1}^{N_e} k_s^j (\epsilon^j)^2 |\bar{\mathbf{e}}^j|. \tag{3}$$

- **Bending** The bending strain for an internal vertex $\mathbf{x}_i$ is defined by the curvature binormal $(\kappa\mathbf{b})_i$, which captures the misalignment between two adjacent edges:

$$(\kappa\mathbf{b})_i = \frac{2\mathbf{t}^{i-1} \times \mathbf{t}^i}{1 + \mathbf{t}^{i-1} \cdot \mathbf{t}^i}. \tag{4}$$

Note that $|(\kappa\mathbf{b})_i| = 2\tan(\phi_i/2)$, where $\phi_i$ is the bending angle between consecutive edges. The material curvature at an interior vertex is defined as

$$\boldsymbol{\kappa}_i = \frac{1}{2} \sum_{j=i-1}^{i} \left( (\kappa\mathbf{b})_i \cdot \mathbf{d}_2^j, (\kappa\mathbf{b})_i \cdot \mathbf{d}_1^j \right)^\top. \tag{5}$$

With these properties defined, we can formulate the bending energy as:

$$U_b = \frac{1}{2} \sum_{i=2}^{N_v-1} \frac{1}{\bar{l}_i} (\boldsymbol{\kappa}_i - \bar{\boldsymbol{\kappa}}_i)^\top B_i (\boldsymbol{\kappa}_i - \bar{\boldsymbol{\kappa}}_i), \tag{6}$$

where $\bar{l}_i = (|\bar{\mathbf{e}}^{i-1}| + |\bar{\mathbf{e}}^i|)/2$ is the Voronoi length for a vertex, $\bar{\boldsymbol{\kappa}}_i$ is the undeformed curvature,

$$B_i = \frac{EA_i}{4} \begin{pmatrix} r_i^2 & 0 \\ 0 & r_i^2 \end{pmatrix}, \tag{7}$$

is the bending stiffness, $E$ is the bending modulus.

- **Twisting** The twisting strain is defined as:

$$\tau_i = m_i - \bar{m}_i, \tag{8}$$

where $m_i = \theta^i - \theta^{i-1} + \underline{m}_i$, $\bar{m}_i$ is the undeformed twist, and $\underline{m}_i$ is the reference twist. Given the twisting stiffness $\beta_i = GA_i r_i^2/2$, where $G$ is the twisting modulus, the twisting energy writes as:

$$U_t = \frac{1}{2} \sum_{i=2}^{N_v-1} \beta_i \frac{(\tau_i)^2}{\bar{l}_i}. \tag{9}$$

**Inextensibility Constraint** In addition, when inextensibility is desired, we can deactivate the stretching energy and use the following geometry projections to enforce the constraint. This approach directly modifies vertex positions to satisfy length constraints, offering greater stability than high-stiffness penalty forces. For each edge $\mathbf{e}^j$ connecting vertices $\mathbf{x}_j$ and $\mathbf{x}_{j+1}$, we first evaluate the constraint function $C(\mathbf{x}_j, \mathbf{x}_{j+1}) = |\mathbf{x}_{j+1} - \mathbf{x}_j| - |\bar{\mathbf{e}}^j|$, where $|\bar{\mathbf{e}}^j|$ is the rest edge length. A correction vector $\Delta\mathbf{x}$ is then calculated for each vertex, weighted by its inverse mass $w_i = 1/m_i$. The correction is distributed along the tangent vector $\mathbf{t}^j$, as shown in the following equations:

$$\begin{aligned} \lambda &= \frac{C(\mathbf{x}_j, \mathbf{x}_{j+1})}{w_j + w_{j+1}}, \\ \Delta\mathbf{x}_j &= \lambda w_j \cdot \mathbf{t}^j, \\ \Delta\mathbf{x}_{j+1} &= -\lambda w_{j+1} \cdot \mathbf{t}^j. \end{aligned} \tag{10}$$

Finally, the vertex positions are updated via $\mathbf{x}_j = \mathbf{x}_j + \Delta\mathbf{x}_j$ and $\mathbf{x}_{j+1} = \mathbf{x}_{j+1} + \Delta\mathbf{x}_{j+1}$. This process is repeated for $N_C$ iterations to enforce the inextensibility of the DLO robustly.

**Bending Plasticity** To simulate elastoplastic materials such as metal wires that can be permanently deformed, we keep track of the rest curvature $\bar{\boldsymbol{\kappa}}_i$ for each internal vertex $i$. Plastic deformation occurs when the magnitude of the elastic curvature $\boldsymbol{\kappa}_i^{el} = \boldsymbol{\kappa}_i - \bar{\boldsymbol{\kappa}}_i$ exceeds a predefined yield threshold, $\sigma_y$. If $|\boldsymbol{\kappa}_i^{el}| > \sigma_y$, the rest curvature is updated according to a creep model, which drives the rest curvature towards the current curvature at a rate proportional to the excess strain. The change in rest curvature, $\Delta\bar{\boldsymbol{\kappa}}_i$, for a given timestep is calculated as:

$$\Delta\bar{\boldsymbol{\kappa}}_i = r_c \cdot \frac{|\boldsymbol{\kappa}_i^{el}| - \sigma_y}{|\boldsymbol{\kappa}_i^{el}|} \boldsymbol{\kappa}_i^{el}, \tag{11}$$

where $r_c$ is the plastic creep rate. This update is then integrated over time, $\bar{\boldsymbol{\kappa}}_i = \bar{\boldsymbol{\kappa}}_i + \Delta\bar{\boldsymbol{\kappa}}_i$, allowing the rod to retain a new intrinsic shape after significant bending.

**Loop Topology**  To properly initialize the material frames for a DLO with a closed-loop topology, we must account for the geometric phase, or holonomy, that arises from parallel transport around a non-trivial curve. A naive sequential transport of the material frame from the first edge ($j = 1$) to the last ($j = N_e$) results in a discontinuity, as parallel transporting the final frame $\bar{\mathbf{d}}_1^{N_e}$ back across the closing edge to the tangent of the first edge, $\mathbf{t}^1$, will not align with the original frame $\bar{\mathbf{d}}_1^1$. To resolve this, we first compute the total holonomy angle, $\psi_H$, which represents this angular mismatch. We then distribute this error evenly across all edges. For each edge $j$, we compute a local correction angle $\phi^j = -\psi_H \cdot (j - 1)/N_e$. The initial, uncorrected material frame vectors, denoted as $\hat{\mathbf{d}}_1^j$ and $\hat{\mathbf{d}}_2^j$, are then rotated by $\phi^j$ to obtain the final, continuous reference frames for the closed loop:

$$\begin{bmatrix} \bar{\mathbf{d}}_1^j \\ \bar{\mathbf{d}}_2^j \end{bmatrix} = \begin{bmatrix} \cos \phi^j & \sin \phi^j \\ -\sin \phi^j & \cos \phi^j \end{bmatrix} \begin{bmatrix} \hat{\mathbf{d}}_1^j \\ \hat{\mathbf{d}}_2^j \end{bmatrix}. \tag{12}$$

This procedure ensures that the material frame is consistent and continuous across the entire loop at its rest state.

**Self-collision and Friction**  To handle collisions and frictional contact among DLOs, we employ a hybrid PBD approach that combines geometric projections for non-penetration with velocity-level updates for friction. We model each segment as a continuous cylinder with radius $r_i$. For any pair of potentially colliding edges, defined by vertices $(\mathbf{x}_i, \mathbf{x}_{i+1})$ and $(\mathbf{x}_j, \mathbf{x}_{j+1})$, we first compute the closest points between the two line segments. These points, $\mathbf{c}_a$ and $\mathbf{c}_b$, are found using barycentric coordinates $t$ and $u \in [0, 1]$ such that $\mathbf{c}_a = (1 - t)\mathbf{x}_i + t\mathbf{x}_{i+1}$ and $\mathbf{c}_b = (1 - u)\mathbf{x}_j + u\mathbf{x}_{j+1}$. The non-penetration constraint is enforced if the penetration depth $d = (r_a + r_b) - |\mathbf{c}_a - \mathbf{c}_b|$ is positive. A correction vector is then applied to the four defining vertices, weighted by their respective inverse masses and barycentric coordinates. The correction for vertex $\mathbf{x}_i$, for example, is calculated as:

$$\Delta \mathbf{x}_i = \frac{d \cdot w_i(1 - t)}{w_i(1 - t)^2 + w_{i+1}t^2 + w_j(1 - u)^2 + w_{j+1}u^2} \cdot \mathbf{n}, \tag{13}$$

where $\mathbf{n} = (\mathbf{c}_a - \mathbf{c}_b)/|\mathbf{c}_a - \mathbf{c}_b|$ is the contact normal. We also repeat the geometric projections for $N_C$ iterations. After the positional correction, friction is resolved at the velocity level. We implement a Coulomb friction model where the tangential velocity update $\Delta \mathbf{v}_t$ is proportional to the normal impulse magnitude, which is approximated from the penetration depth $d$. This velocity correction is then distributed to the four vertices, similarly weighted by their inverse masses and barycentric coordinates, to provide a stable and physically plausible contact response.

## A.2  GRADIENT CHECKPOINTING

A key challenge in achieving full differentiability in the simulation is enabling gradient flow throughout time. A naive solution is to maintain a record of the simulation states at every timestep. However, given that many tasks in DLO-Lab consist of thousands of simulation steps, it becomes unfeasible to store the entire computation graph within the GPU memory. To address this issue, we gained inspiration from FluidLab (Xian et al., 2023) and implemented gradient checkpointing to enable gradient back-propagation along the whole simulation trajectory. Specifically, to allow backpropagation through simulations of arbitrary length without being limited by GPU memory, we utilize a gradient checkpointing technique. During the forward pass, we compute the simulation trajectory in segments. At the end of each segment, we cache a single state checkpoint in CPU memory and discard the intermediate states from the GPU. For the backward pass, we iterate through these checkpoints in reverse order. For each checkpoint, we rerun the forward simulation for that local segment to reconstruct the necessary computational graph, and then perform the backward step. This approach balances the memory cost of storing the entire simulation history with the computational cost of recomputing small portions of the trajectory, allowing the memory requirement to remain independent of the simulation horizon.

## A.3  RENDERING

We utilize the customized LuisaRender (Zheng et al., 2022) available in Genesis to create photorealistic images of our simulation results. The enhanced photorealism enables our simulation environment to generate perception training data for policy learning in DLO manipulation, paving the way for future applications.

| Tasks | #Vertices | Stretching Modulus | Bending Modulus | Twisting Modulus | Inextensibility | Plasticity |
|---|---|---|---|---|---|---|
| Wiring-ring | 60 | – | 1e3 | 1e3 | ✓ | |
| Wiring-post | 60 | – | 1e3 | 1e3 | ✓ | |
| Gathering | 45 | 1e5 | 1e3 | 1e3 | | |
| Coiling | 60 | – | 1e3 | 1e3 | ✓ | |
| Wire Art | 45 | – | 1e4 | – | ✓ | ✓ |
| Slingshot | 12 | 8e5 | 1e5 | – | | |
| Lifting | 30 | – | 5e3 | 1e3 | ✓ | |
| Separation | 60 | – | 5e3 | 1e3 | ✓ | |
| Wrapping | 50 | 1e5 | 1e4 | – | | |

Table 3: **Details of simulation parameters used in DLO-Lab.**

## B  ENVIRONMENTAL SETUP

### B.1  TASK SETUP

In all our manipulation tasks, we use a simulation time step of 2e-4 seconds, and each environment step consists of 5 time steps. We report the details of parameter settings for each task in Table 3.

### B.2  REINFORCEMENT LEARNING SETUP

We use open-source implementations (D'Eramo et al., 2021) of PPO (Schulman et al., 2017) and SAC (Haarnoja et al., 2018) in our environments. We set the batch size to be $2,048$ for both PPO and SAC. Below, we provide other hyperparameters in Table 4-5. Note that $\mathcal{A}$ in Table 5 denotes the dimension of the action space.

| actor lr | 5e-4 |
|---|---|
| actor mlp | [256, 128, 64] |
| critic lr | 5e-4 |
| critic mlp | [256, 128, 64] |
| grad clip | 0.5 |
| entropy coeff. | 5e-3 |

Table 4: **PPO parameters.**

| actor lr | 5e-4 |
|---|---|
| actor mlp | [256, 128, 64] |
| critic lr | 5e-4 |
| critic mlp | [256, 256, 256] |
| grad clip | 0.5 |
| replay buffer | 2e5 |
| target entropy | $-\dim(\mathcal{A})/2$ |

Table 5: **SAC parameters.**

| pop size | 200 |
|---|---|
| elitist | True |
| init sigma | 0.005 |

Table 6: **CMA-ES parameters.**

### B.3  TRAJECTORY OPTIMIZATION SETUP

We use open-source implementations (Hansen et al., 2019) of CMA-ES in our environments. We list the hyperparameters we used in Table 6. For gradient descent, we use the Adam optimizer (Kingma, 2014) with an initial learning rate of 1e-3, which decreases to 1e-6 using a cosine schedule.

## C  SIMULATION PERFORMANCE

We benchmark the running time of our simulator on each task in Table 7. Note that each environment step corresponds to 0.001 seconds.

We further examine how well our simulator aligns with fundamental conservation laws, focusing specifically on the conservation of momentum during two distinct contact scenarios, as illustrated in Figure 6. The scenarios we simulate are "Parallel Contact" and "Inclined Contact." The corresponding plots on the right-hand side track the system's total momentum error over time. It is observed that the error magnitude ranges from approximately 1e-13 to 1e-14, indicating that our simulator effectively conserves momentum during these contact events.

| Tasks | Forward | Forward + Backward |
|---|---|---|
| Wiring-ring | 28.15 ms (36 FPS) | 55.70 ms (18 FPS) |
| Wiring-post | 24.42 ms (41 FPS) | 57.70 ms (17 FPS) |
| Gathering | 24.77 ms (40 FPS) | – |
| Coiling | 23.60 ms (42 FPS) | 56.26 ms (18 FPS) |
| Wire Art | 20.10 ms (50 FPS) | 48.48 ms (21 FPS) |
| Slingshot | 21.98 ms (45 FPS) | – |
| Lifting | 26.72 ms (37 FPS) | – |
| Separation | 22.62 ms (44 FPS) | 54.00 ms (19 FPS) |
| Wrapping | 22.78 ms (44 FPS) | 58.82 ms (17 FPS) |

Table 7: **Running time on an NVIDIA L40s GPU.** We show the average time for a single forward or forward + backpropagation environment step for each task.

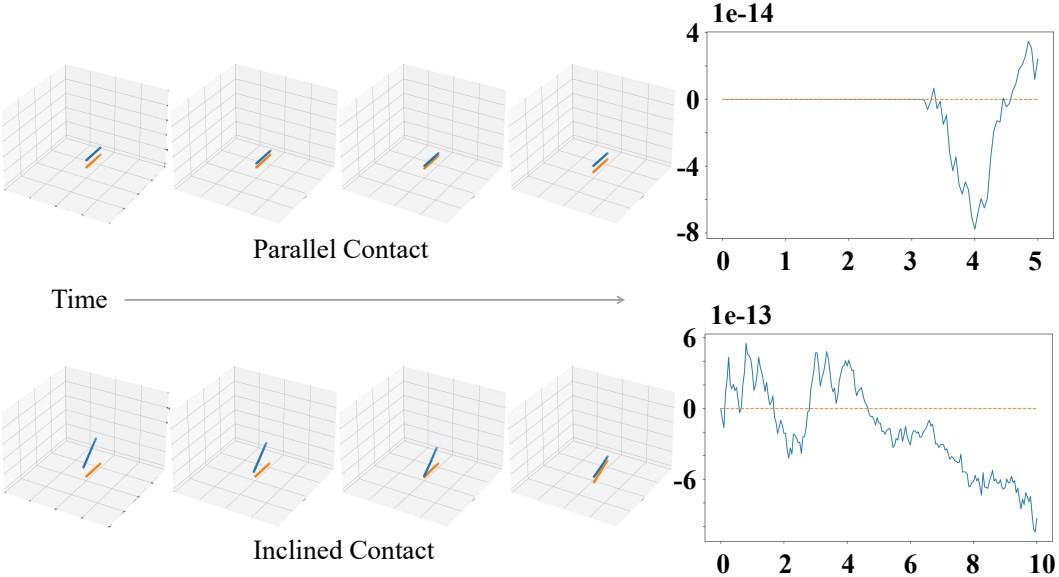

Figure 6: **Analysis of momentum conservation.** On the left, we display the contact cases we considered. On the right, we plot the system's total momentum error over time. It is observed that our simulator conserves momentum with very low errors during contact events.

## D   MORE DETAILS ON AGENT TASK DECOMPOSITION

As introduced in Section 4.4, the proposed agent task decomposition acts as a high-level planner that performs two main functions: (1) it identifies the initial grasping point for each robot, and (2) it breaks down a manipulation task into several subtasks when a change in grasping points is necessary (e.g., **Wiring-ring**). Below, we provide the prompts used in our experiment for the two scenarios. We utilized the Gemini-2.5-Pro (Comanici et al., 2025) model as the VLM agent.

### D.1   IDENTIFYING INITIAL GRASPING POINTS

Below is a sample prompt we used to ask the VLM agent to generate the initial grasping points for each manipulation task.

```
You are an intelligent AI assistant for robotics, physical simulation, and deformable
object manipulation.

Follow the user's requirements carefully and make sure you understand them.

Keep your answers short and to the point.

Do not provide any information that is not required.
```

```
You will suggest initial grasping points for solving various manipulation tasks involving
deformable linear objects, ensuring they are both efficient and robust for the task.

The task is [TASK DESCRIPTION]. The reward for this task is defined as [REWARD DEFINITION].
The rendering of the initial state is provided in the attached image. The [ROPE COLOR] rope
is named ''rope-1''. (The [ROPE COLOR] rope is named ''rope-2''.) The markers shown in the
image indicate the starting point of the linear object. [DESCRIPTION OF THE MAPPING FROM
GRASPING POINT ID TO ROBOT ARM ID].

Using the information provided and your understanding of deformable manipulation, please
recommend initial grasping points for each robot involved in the task. The initial grasping
point should be represented as a number within the range of [0, 1], indicating its position
along the linear object.

Please provide your answer in the following JSON format:
```
{
  "grasping_point_robot_1": ... # some value in [0,1],
  "grasping_point_robot_2": ... # some value in [0,1],
  ...
  "reasoning for the suggestion": ...
}
'''
The output should **only** contain the dictionary.
```

## D.2 TASK DECOMPOSITION

Below is a sample prompt we used to ask the VLM agent to break the task into subtasks when a switch in grasping points is required to complete it.

```
You are an intelligent AI assistant for robotics, physical simulation, and deformable
object manipulation.

Follow the user's requirements carefully and make sure you understand them.

Keep your answers short and to the point.

Do not provide any information that is not required.

You will divide a manipulation task for deformable linear objects into the fewest possible
subtasks, keeping the robot gripper's grasping point on the linear object unchanged in each
subtask. That is, break down the task into subtasks when a switch in grasping points is
required to complete it.

The task is [TASK DESCRIPTION]. The reward for this task is defined as [REWARD DEFINITION].
The rendering of the initial state is provided in the attached image. The [ROPE COLOR] rope
is named ''rope-1''. (The [ROPE COLOR] rope is named ''rope-2''.) The markers shown in the
image indicate the starting point of the linear object. [DESCRIPTION OF THE MAPPING FROM
GRASPING POINT ID TO ROBOT ARM ID].

Using the information provided and your understanding of deformable manipulation, please
recommend a task decomposition.

Please provide your answer in the following JSON format:
```
{
  "subtask_1": ... # concise description of subtask 1
  "subtask_2": ... # concise description of subtask 2
  ...
}
'''
The output should **only** contain the dictionary.
```

After breaking down the task, we begin training the algorithm on the first subtask. We utilize the prompt in Section D.1 to determine the initial grasping points for this subtask. After completing the training, we assess whether the subtask was successful using the agent. If the outcome is positive, we move on to the next subtasks. If the result is negative, we halt the training process.

