# OpenReview forum: "Deformable Linear Object Manipulations with Differentiable Physics"
_ICLR.cc/2026/Conference — Submitted to ICLR 2026_

### Official Review · Reviewer_ind7 · 2025-10-24

**Soundness:** 2
**Presentation:** 2
**Contribution:** 2
**Rating:** 2
**Confidence:** 4

**Summary:**

The paper presents DLO-Lab, a differentiable simulator, and nine benchmark tasks. In the evaluation, the authors tried out PPO/SAC, CMA-ES, and gradient-based trajectory optimization on their benchmarks. A single slingshot sim-to-real example is also shown. In addition, the authors adopt an LLM “agent” to decompose tasks into subtasks and outputs grasp points as re-planning after each subtask.

**Strengths:**

- **Task suite.** The tasks cover a wide range of linear deformable manipulation (routing, wrapping, separating, slingshot). Short descriptions of rewards for each task are provided.
- **Method coverage.** RL (PPO/SAC) and planning-based methods (CMA-ES, gradient-based trajectory optimization) are evaluated.

**Weaknesses:**

- **Task difficulty/coverage is unclear.**
For RL, generalizability is important, but the paper does not show it in the task design. It is unclear how results change under large or medium randomized initial states and targets (e.g., different ball/box placements and angles in Slingshot). The main text points to an appendix for setups, but the results include no robustness evaluation across diverse configurations.

- **Sim-to-real details are insufficient.**
The paper claims the sim-to-real gap is “manageable” based on one Slingshot demo, but there is no description of the system-ID procedure and no statistics on hardware success. Table 3 lists simulation parameters (e.g., slingshot stretching 8e5, bending 1e5), but the paper does not explain how these values were obtained or validated against the real setup. This limits sientific value and reusability.

- **Experiment setup details are missing.**
Nowadays, simulator speed matters, especially for RL. With a massively parallel setup, one can expect PPO to solve these tasks. The problem is that, for deformables, it is not trivial to run as many parallel environments as rigid-body cases. The paper should report #parallel envs, step time/FPS, and wall-clock. These numbers are also essential for the planning baselines to judge how fast (or real-time) they can produce trajectories.

- **Baselines are outdated**
Deformable manipulation has progressed recently in both model-free RL (HEPi, Hoang et al., ICLR 2025) and model-based methods with simulation gradients (SAPO, Xing et al., ICLR 2025). The authors evaluate only outdated baselines.

- **LLM agent usage is under-specified.**
Section 4.4 says the agent decomposes the task and outputs grasp points, re-planning after each subtask. However, the paper does not state whether the authors train or use one controller per task or different controllers per subtask, whether subtask-specific rewards are used, or whether the LLM agent is called during RL training rollouts or only at evaluation.

- **Videos for both success and failure cases**.
For robotic manipulation, videos are necessary to understand what “success” looks like and to judge jerkiness and/or stability. The paper unfortunately does not provide them.

-------------------------------

**Minor points:**

**Reporting**: Table 2 shows “maximum reward within a fixed number of episodes” with mean ± std, but the number of seeds and exact episode budgets per method are not stated. This weakens interpretability and reproducibility.

**Metrics**: heavy emphasis on reward; success rates would provide better intuition.

-------------------------------

(Hoang et al. 2025). Geometry-aware RL for manipulation of varying shapes and deformable objects. ICLR 2025

(Xing et. al. 2025). Stabilizing Reinforcement Learning in Differentiable Multiphysics Simulation. ICLR 2025

**Questions:**

See weaknesses.

---

> ### Author Response · Authors · 2025-11-30
> **Response to Reviewer ind7 1**
>
> We sincerely appreciate the reviewers' comments and try to address them below.
>
> > W1: Task difficulty/coverage is unclear. For RL, generalizability is important, but the paper does not show it in the task design. It is unclear how results change under large or medium randomized initial states and targets (e.g., different ball/box placements and angles in Slingshot). The main text points to an appendix for setups, but the results include no robustness evaluation across diverse configurations.
>
> We have provided additional details about the experiment setups in Section B. For robustness evaluation, we tested the PPO algorithm on the "Separation" task and shared the video results on the static page added to the supplmentary material (see "Robustness Evaluation"). Specifically, we altered the initial positions, stiffness, thickness, and topology of the DLOs and re-evaluated the algorithm's performance.
>
>
> > W2: Sim-to-real details are insufficient. The paper claims the sim-to-real gap is “manageable” based on one Slingshot demo, but there is no description of the system-ID procedure and no statistics on hardware success. Table 3 lists simulation parameters (e.g., slingshot stretching 8e5, bending 1e5), but the paper does not explain how these values were obtained or validated against the real setup. This limits sientific value and reusability.
>
> In our sim-to-real transfer experiment, we empirically tune the material parameters to align with real-world videos. In the future, we plan to set up real-to-sim environments and conduct system identification using differentiable simulation and rendering techniques.
>
> > W3: Experiment setup details are missing. Nowadays, simulator speed matters, especially for RL. With a massively parallel setup, one can expect PPO to solve these tasks. The problem is that, for deformables, it is not trivial to run as many parallel environments as rigid-body cases. The paper should report #parallel envs, step time/FPS, and wall-clock. These numbers are also essential for the planning baselines to judge how fast (or real-time) they can produce trajectories.
>
> We have supplemented the running time analysis in Section B of the manuscript and show the results below. Specifically, we report the average time for a single forward or forward + backpropagation environment step for each task. Note that one environment step corresponds to 0.001 seconds. We use 100 parallel environments for training the RL and CMA-ES algorithms.
>
> | Tasks | Forward | Forward + Backward |
> | :--- | :---: | :---: |
> | **Wiring-ring** | 28.15 ms (36 FPS) | 55.70 ms (18 FPS) |
> | **Wiring-post** | 24.42 ms (41 FPS) | 57.70 ms (17 FPS) |
> | **Gathering** | 24.77 ms (40 FPS) | – |
> | **Coiling** | 23.60 ms (42 FPS) | 56.26 ms (18 FPS) |
> | **Wire Art** | 20.10 ms (50 FPS) | 48.48 ms (21 FPS) |
> | **Slingshot** | 21.98 ms (45 FPS) | – |
> | **Lifting** | 26.72 ms (37 FPS) | – |
> | **Separation** | 22.62 ms (44 FPS) | 54.00 ms (19 FPS) |
> | **Wrapping** | 22.78 ms (44 FPS) | 58.82 ms (17 FPS) |
>
> > W4: Baselines are outdated. Deformable manipulation has progressed recently in both model-free RL (HEPi, Hoang et al., ICLR 2025) and model-based methods with simulation gradients (SAPO, Xing et al., ICLR 2025). The authors evaluate only outdated baselines.
>
> We appreciate the suggestion to include more recent baselines. Unfortunately, we were unable to complete the experiments for these two baselines during the rebuttal period. We will update these results once they are done.
>
> > W5: LLM agent usage is under-specified. Section 4.4 says the agent decomposes the task and outputs grasp points, re-planning after each subtask. However, the paper does not state whether the authors train or use one controller per task or different controllers per subtask, whether subtask-specific rewards are used, or whether the LLM agent is called during RL training rollouts or only at evaluation.
>
> In our experiment, we prompt the agent to identify the initial grasping points for each task based on the task description and the rendered images of the initial states. We then use these initial grasping points for different algorithms. When task decomposition is necessary, as in the case of the Wiring-ring task, we first instruct the agent to break the task down into several subtasks according to the requirements for switching grasping points.  During training, we begin by training the algorithm on the first subtask and utilize the agent to evaluate whether this task is deemed successful. If the outcome is positive, we proceed to the subsequent subtasks. If not, the training process is halted.
>
> We provided more details on the implementation of the proposed agent task decomposition framework in Section D of the manuscript.

---

> ### Author Response · Authors · 2025-11-30
> **Response to Reviewer ind7 2**
>
> > W6: Videos for both success and failure cases. For robotic manipulation, videos are necessary to understand what “success” looks like and to judge jerkiness and/or stability. The paper unfortunately does not provide them.
>
> We have included video results for one RL baseline (PPO) and one trajectory optimization baseline (CMA-ES) on the static page in the supplementary material.
>
> > Reporting: Table 2 shows “maximum reward within a fixed number of episodes” with mean ± std, but the number of seeds and exact episode budgets per method are not stated. This weakens interpretability and reproducibility.
>
> We experimented with three random seeds for each method and report the mean and standard deviation in Table 2. Each method was trained for 20 episodes, with each episode comprising 2,000 environment steps.
>
> > Metrics: heavy emphasis on reward; success rates would provide better intuition.
>
> Many tasks in our benchmark lack a clear distinction between success and failure. Our goal is to provide an objective measurement for various algorithms. For a more intuitive understanding, we have included some video results on the static page in the supplementary material.

---

### Official Review · Reviewer_jgdN · 2025-10-31

**Soundness:** 2
**Presentation:** 2
**Contribution:** 2
**Rating:** 4
**Confidence:** 5

**Summary:**

The paper presents a suite of deformable linear object manipulations asks, which are built with differentiable simulation Genesis. The main difference comparing to peer works lies in the support of plastic deformation and loop topologies. The proposed tasks cover a range of skills of wiring, coiling, wrapping a linear deformable object against some ambient rigid bodies. Reward design for each task is presented. The paper also includes a decomposition of the multi-stage manipulation task by querying an LLM. The performance with RL and evolution strategy methods are reported. A sim-to-real case is demonstrated on the slingshot task.

**Strengths:**

1. Simulating and benchmarking linear deformable object manipulation is an important topic and has much relevance to machine learning to robotics.
2. The technical details are easy to grasp. An implementation as an integral part of open-sourced genesis simulation is benefiting to open and reproducible research.

**Weaknesses:**

1. Limited scientific novelty. All technical methods are well known and the work seems like an aggregation of them for the application to linear deformable objects.
2. Vague contributions. The paper seems trying to include many points that are distant from the core theme of simulating/benchmarking linear deformable manipulation. What is the purpose of the agentic decomposition from LLM? How does it benefit the task scope that the paper is working towards?
3. Relevance of benchmarking tasks. From a robotics perspective, it is unclear how the proposed task suite is capturing the main challenges/key points in manipulating linear deformable objects. And the tasks appear a bit artificial given it is not hard to find industrial or household cases that have a better resemblance of reality, such as cable plugging or tying shoelaces.
4. Unclear message from the skill learning results. As a benchmark, I suppose the tasks should challenge the existing standard methods while the results in Figure 4 seem that they are already solvable? Are the increasing return curves indicating task successes here? What would the authors expect the users to do with the benchmark if standard methods are already reaching "competitive performance"?

**Questions:**

1. Can the paper make a concise and clear statement about its scientific contribution?
2. What is the relevance of the proposed task suite to real-world scenarios?
3. Can the LLM-based task decomposition and proposed grasping points be shown to be necessary or useful for a benchmark paper?
4. What are the benefits of the proposed task suites comparing to the peers besides the support of "plastic bending" and "loop topology". In the end, neither of the two features look like something unimaginable if the exiting "labs" got some extensions. Why a new "lab" is necessary here? Are there any other arguments from runtime performance and robustness perspectives?

---

> ### Author Response · Authors · 2025-11-30
> **Response to Reviewer jgdN 1**
>
> We are grateful for your constructive comments. Below, we respond to the concerns raised in the review.
>
> > W1: Limited scientific novelty. All technical methods are well known and the work seems like an aggregation of them for the application to linear deformable objects.
>
> Although our simulation implementation is grounded in established mathematical theories, we respectfully disagree with the assertion that our work lacks scientific novelty. The originality of our research lies in the innovative integration and meaningful adaptation of these theories to address specific, open problems in deformable linear object manipulation that existing implementations do not effectively tackle (see Table 1). To the best of our knowledge, our work is the first to support a wide variety of material properties commonly found in everyday DLOs, while being both parallelizable and differentiable for learning robotics skills with these objects.
>
> > W2: Vague contributions. The paper seems trying to include many points that are distant from the core theme of simulating/benchmarking linear deformable manipulation. What is the purpose of the agentic decomposition from LLM? How does it benefit the task scope that the paper is working towards?
>
> The proposed agent task decomposition leverages the unique linear characteristics of DLOs to enhance robotic manipulation involving these objects. Specifically, it serves as a high-level planner that performs two main functions: (1) identifying the initial grasping point for each robot, and (2) breaking down a task into multiple subtasks when it is necessary to change grasping points. For example, in the Wiring-ring task, the grasping point needs to change before and after the ring passes through. In such cases, the agent is prompted to divide the task into several subtasks and determine new grasping points based on the visual renderings of the current state.
>
> > W3: Relevance of benchmarking tasks. From a robotics perspective, it is unclear how the proposed task suite is capturing the main challenges/key points in manipulating linear deformable objects. And the tasks appear a bit artificial given it is not hard to find industrial or household cases that have a better resemblance of reality, such as cable plugging or tying shoelaces.
>
> We want to clarify that our benchmark is specifically designed to assess fundamental manipulation primitives rather than complete end-to-end applications. Our suite aims to break down real-world activities into isolated, testable dimensions of physical complexity that standard benchmarks often overlook.
>
> We have intentionally selected tasks to focus on the physical properties required for both household and industrial scenarios. This "artificial" nature is deliberate, allowing for precise quantitative analysis of these properties. Please refer to the table below, which links our designed tasks with real-world activities.
>
> |Tasks| Physical Challenge| Real-world Equivalent|
> |:---|:---|:---|
> |**Wiring-ring/post**| Precision insertion & Path following | Cable wiring / PCB assembly|
> |**Wire Art**| Plasticity & Shape retention | Wire forming |
> |**Slingshot**| High stretching and bending stiffness | Slingshot game|
> |**Separation**| Continuous friction & Entanglement | Knot Untying |
> |**Wrapping**| Loop topology | Rubber band assemply|
>
> > W4: Unclear message from the skill learning results. As a benchmark, I suppose the tasks should challenge the existing standard methods while the results in Figure 4 seem that they are already solvable? Are the increasing return curves indicating task successes here? What would the authors expect the users to do with the benchmark if standard methods are already reaching "competitive performance"?
>
> While trajectory optimization baselines, i.e., CMA-ES and GD, could obtain a fixed solvable trajectory for most of the tasks, it remains challenging for RL baselines to learn a successful policy for many tasks in our benchmark (e.g., Gathering, Lifting, Slingshot, Wire Art, Wrapping, Wiring-ring and Wiring-post). Please check the video demonstrations for the PPO on the static page we added to the supplementary material.

---

> ### Author Response · Authors · 2025-11-30
> **Response to Reviewer jgdN 2**
>
> > Q1: Can the paper make a concise and clear statement about its scientific contribution?
>
> We summarize our scientific contributions in three key areas:
>
> **Simulation**: We present a novel differentiable physics engine for deformable objects that integrates elasticity, bending plasticity, and two-way rigid/soft body coupling. This innovation overcomes the limitations of previous works, which lacked support for complex material behaviors and interactions involving multiple contacts.
>
> **Benchmarking**: We establish a standardized benchmark suite specifically designed to isolate key physical challenges in DLO manipulation (e.g., topology, friction, shape retention), enabling rigorous algorithmic comparison beyond ad-hoc scenarios.
>
> **Planning**: We propose an agent task decomposition framework that leverages the linear topology of DLOs. This agent breaks down long-horizon manipulation problems into multiple subtasks when it is necessary to switch grasping points and automatically identifies effective grasping points for each subtask.
>
> > Q2: What is the relevance of the proposed task suite to real-world scenarios?
>
> Please refer to the table shown in W3.
>
> > Q3: Can the LLM-based task decomposition and proposed grasping points be shown to be necessary or useful for a benchmark paper?
>
> Without breaking down the task into several subtasks, switching grasping points becomes necessary; tasks like Wiring-ring become intractable for gradient-based methods. With task decomposition in the agent, the learning algorithms can focus on each subtask's specific goal.
>
> > Q4: What are the benefits of the proposed task suites comparing to the peers besides the support of "plastic bending" and "loop topology". In the end, neither of the two features look like something unimaginable if the exiting "labs" got some extensions. Why a new "lab" is necessary here? Are there any other arguments from runtime performance and robustness perspectives?
>
> The support of a wider range of DLO properties allows us to design diverse tasks that existing simulators may not be able to handle. Moreover, we believe that incorporating these features into current manipulation benchmarks is not just a matter of engineering; it requires a fundamental shift in the simulation paradigm.
>
> Specifically, existing benchmarks for DLOs, such as SoftGym, PlasticineLab, and DaxBench, primarily use Position-Based Dynamics (PBD) or Material Point Method (MPM) as their underlying simulation techniques. To simulate a thin wire that has bending and twisting stiffness using MPM, a very high resolution must be employed to prevent numerical fracture. In contrast, our suite is based on discrete elastic rod theory, which simplifies the state space from a 3D volume to a 1D centerline with orientation frames. This approach results in simulation speeds for slender objects that are orders of magnitude faster than volumetric methods.
>
> Additionally, we support the coupling of DLOs with both rigid and soft bodies, enabling the use of robot arms equipped with grippers to perform manipulation tasks. Unlike previous benchmarks such as PlasticineLab, Thin-shellLab, and DaxBench, which assume flying grippers and are therefore less realistic, our framework better represents real-world scenarios.

---

### Official Review · Reviewer_GzRa · 2025-11-01

**Soundness:** 2
**Presentation:** 2
**Contribution:** 2
**Rating:** 2
**Confidence:** 3

**Summary:**

This paper introduces a novel differentiable physics simulator for Deformable Linear Objects (DLOs) aimed at addressing the limitations of existing simulation environments for robotic manipulation. The primary contribution is a unified framework, "DLO-Lab," that models a wide and comprehensive range of DLO physical properties, including elasticity, inextensibility, bending plasticity, and loop topologies. They propose a benchmark suite of nine diverse DLO manipulation tasks designed to test these varied physical properties and interactions. The paper evaluates the performance of different policy learning paradigms on this benchmark: reinforcement learning (PPO, SAC), sampling-based trajectory optimization (CMA-ES), and gradient-based trajectory optimization. The results show that gradient-based methods are the most sample-efficient when gradients are available, while CMA-ES is a more robust general-purpose optimizer, particularly for tasks with sparse contact.

**Strengths:**

The framework incorporates several features that were not realized at the same time with other simulators, for example, full differentiability with a rich set of physical properties, bending plasticity, and loop topology, as well as coupling with both rigid and other soft-body materials (MPM), as shown in Table 1.


For the paper clarity, the motivation is strong and well-articulated. Visual aids like Table 1 (comparison to prior work) and Figure 2 (task illustrations) are highly effective at communicating the paper's contributions and scope.

**Weaknesses:**

1) Sim-to-Real evaluation is limited

The real-world experiment (Sec 5.3) can be a sound proof of concept, but the results shown in the paper are very limited. It demonstrates the transfer of a single open-loop trajectory for one task, which validates the simulator's kinematics and some dynamics but does not validate its suitability for closed-loop control, where a policy would need to react to perception feedback. Inaccuracies in simulated contact forces or friction models, for example, would only be exposed in a closed-loop setting. Additionally, we would like to see the comparison between the simulator and the real in the video.

2) Disconnection of agent task decomposition

Section 4.4 introduces the LLM-based agent for task decomposition, which seems disconnected from the paper's core contribution (the simulator and benchmark).. It's an application of the benchmark, but it's presented as a key feature of the simulator. The implementation details are missing from the text, for example, the details of models, prompt engineering, and robustness.


3) Low reproducibility of the results

The paper does not provide the code or much information about the implementation details, which makes it hard for readers to utilize the proposed framework or reproduce results.

**Questions:**

1) How robust is the framework the paper proposes? I would like to see the robustness of changing physical parameters, for example, transferring policies for more contact-heavy tasks, or tasks where friction dynamics are more critical.

2) Regarding the Agent Task Decomposition (Sec 4.4), what is the precise role of this agent? Is it used to generate the trajectories for the benchmarked algorithms (PPO, CMA-ES, GD)? Or is it a separate, high-level planner that would use policies trained by these methods as sub-skills?

---

> ### Author Response · Authors · 2025-11-30
> **Response to Reviewer GzRa**
>
> Thank you for reviewing our work. We provide the responses to your concerns below.
>
> > W1: Sim-to-Real evaluation is limited
>
> We appreciate the reviewer’s suggestion. Due to the time constraints of the rebuttal period, we are unable to fully implement closed-loop policies in the real world at this time. However, we will continue to update our results. Regarding the original sim-to-real transfer experiment, we have included a video in the supplementary material. Please refer to "page/index.html" in the uploaded zip file for more information.
>
>
> > W2: Disconnection of agent task decomposition
>
> The agent task decomposition is more than just an application of the benchmark; it is designed to enhance robot manipulation of deformable linear objects by leveraging their linear characteristics. With this proposed agent, robot grippers can identify suitable grasping points to address various tasks. For tasks that require switching between grasping points, the agent is essential for breaking down the task into several subtasks, guiding the robot on which grasping points to use upon completing each subtask.
>
> We also provided additional implementation details for the agent in Section D of the manuscript.
>
> > W3: Low reproducibility of the results
>
> We presented the implementation details in Section A.1 of the manuscript. Additionally, we included more information about the experimental setup in Section B of the manuscript. Our code will be made open-source to ensure reproducibility.
>
> > Q1: How robust is the framework the paper proposes? I would like to see the robustness of changing physical parameters, for example, transferring policies for more contact-heavy tasks, or tasks where friction dynamics are more critical.
>
> We have added more video demonstrations using our simulator to the static page included in the supplementary material.  We also evaluate the robustness of a trained policy by changing physical parameters (e.g., initial locations, stiffness, and thickness) and presented the result in "Robustness Evaluation" section on the static page.
>
> > Q2: Regarding the Agent Task Decomposition (Sec 4.4), what is the precise role of this agent? Is it used to generate the trajectories for the benchmarked algorithms (PPO, CMA-ES, GD)? Or is it a separate, high-level planner that would use policies trained by these methods as sub-skills?
>
> The proposed agent task decomposition functions as a high-level planner that performs two main tasks: (1) it identifies the initial grasping point for each robot, and (2) it breaks down a task into several subtasks when a change in grasping points is necessary. For example, in the Wiring-ring task, the grasping point must change before and after the ring passes through. In such instances, the agent is prompted to divide the task into multiple subtasks and identify new grasping points based on the renderings of the current state.

---

### Official Review · Reviewer_Fuxw · 2025-11-09

**Soundness:** 2
**Presentation:** 3
**Contribution:** 2
**Rating:** 4
**Confidence:** 3

**Summary:**

This paper focuses on manipulation of deformable linear objects (DLOs), such as ropes and cables. The authors introduce a differentiable physics simulator tailored for DLOs, enabling gradient-based optimization for planning and learning. The simulator models elastic and frictional behaviors through a differentiable mass-spring representation, making it suitable for both analytical gradient computation and policy training.

The authors also present a benchmark suite of DLO manipulation tasks, including rope straightening, knot tying, threading, and shape matching. The benchmark systematically evaluates several representative policy learning paradigms, including reinforcement learning (RL), trajectory optimization, and sampling-based motion planning. The evaluation highlights the limitations of existing approaches and motivates future algorithmic innovations for deformable object manipulation.

**Strengths:**

(+) The introduction of a differentiable simulator for DLOs is highly valuable for the community. It provides a reproducible and extensible software for studying deformable object control, a domain that has historically lacked standardized tools.

(+) The authors evaluate a wide spectrum of methods (RL, trajectory optimization, and sampling-based approaches) under a unified environment, offering an insightful comparison of their respective strengths and weaknesses.

(+) By leveraging a differentiable mass-spring formulation, the simulator enables gradient-based learning and efficient optimization, a clear improvement over prior non-differentiable simulators like PyBullet or SOFA.

(+) Well-structured benchmark tasks: The benchmark tasks span a range of difficulty levels and manipulation types, providing a clear gradient for research progression.

**Weaknesses:**

(-) Lack of methodological novelty. The paper’s main contribution lies in the simulator and benchmark design; no new algorithm or modeling technique is proposed beyond existing differentiable physics formulations.

(-) Limited insight into algorithmic takeaways. While multiple learning paradigms are compared, the discussion lacks deeper analysis or clear conclusions about why certain methods succeed or fail, or what principles should guide future work.

(-) Weak real-world validation. Although the simulator is claimed to be physically accurate, real-world experiments are minimal and qualitative. A stronger demonstration of sim-to-real consistency would substantially improve the paper’s credibility.

(-) Scalability and efficiency concerns. The computational cost of differentiable simulation for long DLOs or contact-rich scenarios is not clearly discussed. Real-time feasibility remains uncertain.

(-) Limited generalization across object materials: The experiments are primarily conducted with homogeneous ropes; the simulator’s ability to handle variable stiffness or heterogeneous materials is not tested.

**Questions:**

Is the simulator capable of handling self-collisions or topological changes (e.g., forming or untying knots)?

How well does the differentiable simulator match real-world dynamics? Are there quantitative comparisons between simulated and physical trajectories?

Which policy learning algorithms benefited most from differentiable gradients, and which still relied heavily on sampling?

Could the simulator be extended to sheet-like deformable objects (e.g., cloth) or more complex soft structures?

What is the expected runtime per simulation step, and how does it scale with DLO length and resolution?

---

> ### Author Response · Authors · 2025-11-30
> **Response to Reviewer Fuxw 1**
>
> Thank you for reviewing our work and providing valuable comments. Here are our responses to your concerns.
>
> > W1: Lack of methodological novelty. The paper’s main contribution lies in the simulator and benchmark design; no new algorithm or modeling technique is proposed beyond existing differentiable physics formulations.
>
> We aim to clarify that the primary focus of this work is the introduction of a new simulation engine along with a benchmark for robotics manipulation involving deformable linear objects (DLOs). Therefore, it falls outside our scope to design new techniques for addressing specific manipulation tasks.
>
> Below, we summarize the main contributions of our work:
>
> a) We presented a differentiable simulation engine for deformable linear objects (DLOs) that includes a range of material behaviors, whereas previous work only supported a limited set of features. Our engine also enables interactions between DLOs and both rigid and soft bodies, making this work, to the best of our knowledge, the first of its kind to achieve such capability.
>
> b) We developed a manipulation benchmark specifically designed for learning robotic skills with DLOs. This benchmark includes common tasks such as wiring, coiling, and using a slingshot. Furthermore, the simulation engine allows users to create new learning environments, extending beyond the nine provided scenarios.
>
> c) We proposed an agent task decomposition framework that leverages the linearity characteristics of DLOs to determine appropriate grasping points for robotic grippers. This framework also allows for the division of tasks into several subtasks when it is necessary to switch grasping points.
>
> We believe that all three aspects represent novel contributions to the field of robotic manipulation with deformable linear objects.
>
>
> > W2: Limited insight into algorithmic takeaways. While multiple learning paradigms are compared, the discussion lacks deeper analysis or clear conclusions about why certain methods succeed or fail, or what principles should guide future work.
>
> We revised the discussion to move beyond mere performance comparisons and instead analyze the underlying reasons for the success or failure of each paradigm. Based on our experiments (Section 5.2), we have identified three governing principles:
>
> a) Reinforcement learning (RL) algorithms exhibit significant deficiencies in sample efficiency and require substantially more data to learn effective policies. This is because they must implicitly learn the laws of physics from scratch through trial and error.
>
> b) In contrast, gradient-based trajectory optimization directly utilizes the analytical gradients from the simulator, resulting in significantly improved convergence speed. This indicates that for deformable objects, incorporating physics priors—specifically, gradient information from the differentiable simulator—is much more efficient than black-box exploration. However, the gradient-based method struggles in tasks where the reward depends on the state of other passive objects, such as Gathering and Lifting. In these cases, the gradient flow can be discontinuous; when the deformable object loses contact with other objects, the gradient chain breaks, rendering local optimization challenging.
>
> c) CMA-ES occupies a middle ground. It is robust to the non-smooth energy landscapes caused by contact, where gradient-based methods often fail, because it does not rely on gradients. However, it struggles with high-precision tasks (e.g., Wiring-ring and Wire Art), as random sampling in high-dimensional action spaces rarely identifies the precise, optimal trajectory needed for fine manipulation.
>
> Based on these insights, an interesting direction for future work could involve combining RL algorithms with gradient information. For instance, one could explore Actor-Critic methods where the Critic utilizes the gradients from the differentiable simulator to guide learning, while the Actor addresses the discontinuous nature of contact events.
>
> > W3: Weak real-world validation. Although the simulator is claimed to be physically accurate, real-world experiments are minimal and qualitative. A stronger demonstration of sim-to-real consistency would substantially improve the paper’s credibility.
>
> We appreciate the reviewer’s suggestion regarding real-world validation. While we are making every effort to set up real-world verification, challenges such as sensor noise and occlusion during the collection of accurate pose data for highly deformable objects in dynamic contact have hindered our progress within the limited timeframe of the discussion period. We will continue to update our results. In the meantime, we have included an analysis in Figure 6 of the manuscript that examines how well the simulator aligns with the fundamental laws of physics. Our findings indicate that the simulator conserves momentum with a very low error during different contact events.

---

> ### Author Response · Authors · 2025-11-30
> **Response to Reviewer Fuxw 2**
>
> > W4: Scalability and efficiency concerns. The computational cost of differentiable simulation for long DLOs or contact-rich scenarios is not clearly discussed. Real-time feasibility remains uncertain.
>
> We show the average time for a single forward or forward + backpropagation environment step for each task below. Note that one environment step corresponds to 0.001 seconds.
>
> | Tasks | Forward | Forward + Backward |
> | :--- | :---: | :---: |
> | **Wiring-ring** | 28.15 ms (36 FPS) | 55.70 ms (18 FPS) |
> | **Wiring-post** | 24.42 ms (41 FPS) | 57.70 ms (17 FPS) |
> | **Gathering** | 24.77 ms (40 FPS) | – |
> | **Coiling** | 23.60 ms (42 FPS) | 56.26 ms (18 FPS) |
> | **Wire Art** | 20.10 ms (50 FPS) | 48.48 ms (21 FPS) |
> | **Slingshot** | 21.98 ms (45 FPS) | – |
> | **Lifting** | 26.72 ms (37 FPS) | – |
> | **Separation** | 22.62 ms (44 FPS) | 54.00 ms (19 FPS) |
> | **Wrapping** | 22.78 ms (44 FPS) | 58.82 ms (17 FPS) |
>
> We also test how the running time (FPS) scale with (1) the length of the DLO (with the same resolution) and (2) the resolution of the DLO (with the same length) on the task Coiling below.
>
> a) Length
> | 1x (Original) | 2x | 10x |
> |:---:|:---:|:---:|
> |42 FPS | 42 FPS | 39 FPS |
>
> b) Resolution
> | 1x (Original) | 2x | 10x |
> |:---:|:---:|:---:|
> |42 FPS | 42 FPS | 40 FPS |
>
>
> > W5: Limited generalization across object materials: The experiments are primarily conducted with homogeneous ropes; the simulator’s ability to handle variable stiffness or heterogeneous materials is not tested.
>
>
> We wish to emphasize that our framework fully supports heterogeneous materials by design. As shown in Appendix A.1, the potential energies are computed locally at each stencil (edge/vertex) using spatially varying parameters (e.g., $B_i$ for bending stiffness at vertex $i$). This allows us to model composite structures, such as a cable that transitions from flexible to stiff near a connector. While we restricted our benchmark to homogeneous materials to establish a standardized baseline for robotic manipulation tasks, the simulator is ready for heterogeneous applications without modification.
>
> > Q1: Is the simulator capable of handling self-collisions or topological changes (e.g., forming or untying knots)?
>
> Yes, please refer to the paragraph titled "Self-Collision and Friction" in Section A.1, where we discuss our approach to self-collision and friction in detail.
>
> > Q2: How well does the differentiable simulator match real-world dynamics? Are there quantitative comparisons between simulated and physical trajectories?
>
> Please refer to W3 above.
>
> > Q3: Which policy learning algorithms benefited most from differentiable gradients, and which still relied heavily on sampling?
>
> In our experiment, GD (gradient-descent) leverages gradient information, while other methods (PPO, SAC, and CMA-ES) still rely on sampling.
>
> > Q4: Could the simulator be extended to sheet-like deformable objects (e.g., cloth) or more complex soft structures?
>
> We appreciate the reviewer's insightful question regarding generalizability. Although our current implementation focuses on 1D discrete elastic rods to tackle the specific challenges of DLO manipulation, our framework can be extended to support 2D sheet-like objects, such as cloth, with additional engineering efforts. However, we believe this topic is beyond the scope of this paper, as there are already existing differentiable simulators for thin-shell objects, such as Thin-ShellLab.
>
> > Q5: What is the expected runtime per simulation step, and how does it scale with DLO length and resolution?
>
> Please refer to W4 above.

---

### Author Response · Authors · 2025-11-30
**General Response to All Reviewers**

We sincerely thank all reviewers for their time and effort in reviewing our paper. We greatly appreciate their constructive comments, which are aimed at improving the quality of our work.

We revised our paper to include more implementation details and added a static page to host video results in the supplementary materials. Below is a summary of the updated content:

a) In Section B of the manuscript, we provided additional information about the environmental setups for various algorithms.

b) In Section C, we included more analysis regarding both the efficiency and accuracy of our simulator.

c) In Section D, we added more details about the proposed agent task decomposition framework.

d) In the static page (please check the file "page/index.html" in the zip file we uploaded to the supplementary material), we have included additional video results showcasing:
- Various material behaviors that our simulator supports (teaser)
- Two-way coupling with rigid and soft bodies (teaser)
- Replayed trajectories for each task in our benchmark ("Task Demos")
- Robustness evaluation of the PPO algorithm ("Robustness Evaluation")
- Sim-to-real video demo ("Sim-to-Real Demo")

We hope our responses below will address the reviewers' concerns regarding our work. Additionally, we will make the code open-source to ensure reproducibility. Once again, we thank all reviewers for their time and effort!

---

### Meta-Review · Area_Chair_KRYK · 2026-01-12

**Summary:**

A differentiable physics simulator is developed for deformable objects spanning physical properties such as elasticity, inextensibility, bending plasticity, and loop topologies. The paper comes with nine diverse manipulation tasks designed to test these varied physical properties and interactions. While being an important open topic in Robotics, unfortunately the paper has limited sim2real evals, limited overall novelty, low reproducibility, and somewhat disconnected focus between sim/benchmark & LLM agents.

3 of 4 reviews highlighted sim2real validation weaknesses which are not addressed. For example, the authors acknowledge that "sensor noise and occlusion during the collection of accurate pose data for highly deformable objects in dynamic contact have hindered our progress within the limited timeframe of the discussion". This is a critical weakness.

**Reviewer Concerns:**

The authors engaged closely in the rebuttal period and provided detailed responses.

Reviewer GzRA:  sim2real gap concerns are still outstanding: the authors acknowledge that "Due to the time constraints of the rebuttal period, we are unable to fully implement closed-loop policies in the real world at this time."

Reviewer ind7: baseline results are missing: the authors acknowledge "unable to complete the experiments for these two baselines" in time.

**Reviewer Scores:**

3 of 4 reviews highlighted sim2real validation weaknesses which are not addressed. For example, the authors acknowledge that "sensor noise and occlusion during the collection of accurate pose data for highly deformable objects in dynamic contact have hindered our progress within the limited timeframe of the discussion". This is a critical weakness and hence I do not believe scores would have changed enough to flip the decision.

---

### Decision · Program_Chairs · 2026-01-26

Reject